# Comparative Biomonitoring of Arsenic Exposure in Mothers and Their Neonates in Comarca Lagunera, Mexico

**DOI:** 10.3390/ijerph192316232

**Published:** 2022-12-04

**Authors:** José Javier García Salcedo, Taehyun Roh, Lydia Enith Nava Rivera, Nadia Denys Betancourt Martínez, Pilar Carranza Rosales, María Francisco San Miguel Salazar, Mario Alberto Rivera Guillén, Luis Benjamín Serrano Gallardo, María Soñadora Niño Castañeda, Nacny Elena Guzmán Delgado, Jair Millán Orozco, Natalia Ortega Morales, Javier Morán Martínez

**Affiliations:** 1Departamento de Bioquímica y Farmacología, Centro de Investigaciones Biomédicas, Facultad de Medicina, Universidad Autónoma de Coahuila Torreón, Torreón 27000, Mexico; 2Department of Epidemiology and Biostatistics, School of Public Health, Texas A&M University, College Station, TX 77843, USA; 3Departamento de Biología Celular y Ultraestructura, Centro de Investigaciones Biomédicas, Facultad de Medicina, Universidad Autónoma de Coahuila Torreón, Torreón 27000, Mexico; 4Centro de Investigaciones Biomédicas del Noreste, Instituto Mexicano del Seguro Social, Monterrey 64000, Mexico; 5División de Investigaciones en Salud, Unidad Médica de Alta Especialidad, Hospital de Cardiología #34, Instituto Mexicano del Seguro Social, Monterrey 64000, Mexico; 6Unidad Laguna, Universidad Autónoma Agraria Antonio Narro, Raúl López Sánchez, Torreon 27000, Mexico

**Keywords:** arsenic, drinking water, biomonitoring, breast milk, neonates, pregnancy

## Abstract

Multiple comorbidities related to arsenic exposure through drinking water continue to be public problems worldwide, principally in chronically exposed populations, such as those in the Comarca Lagunera (CL), Mexico. In addition, this relationship could be exacerbated by an early life exposure through the placenta and later through breast milk. This study conducted a comparative analysis of arsenic levels in multiple biological samples from pregnant women and their neonates in the CL and the comparison region, Saltillo. Total arsenic levels in placenta, breast milk, blood, and urine were measured in pregnant women and their neonates from rural areas of seven municipalities of the CL using atomic absorption spectrophotometry with hydride generation methodology. The average concentrations of tAs in drinking water were 47.7 µg/L and 0.05 µg/L in the exposed and non-exposed areas, respectively. Mean levels of tAs were 7.80 µg/kg, 77.04 µg/g-Cr, and 4.30 µg/L in placenta, blood, urine, and breast milk, respectively, in mothers, and 107.92 µg/g-Cr in neonates in the exposed group, which were significantly higher than those in the non-exposed area. High levels of urinary arsenic in neonates were maintained 4 days after birth, demonstrating an early arsenic exposure route through the placenta and breast milk. In addition, our study suggested that breastfeeding may reduce arsenic exposure in infants in arsenic-contaminated areas. Further studies are necessary to follow up on comorbidities later in life in neonates and to provide interventions in this region.

## 1. Introduction

Exposures to arsenic (As) can be assessed through different sources: contaminated food intake, cigarette consumption and occupational exposure, the use of certain cosmetics, and consumption through drinking water [1,2,3,4,5]. There are two types of arsenic, organic and inorganic arsenic. Organic arsenic such as arsenobetaine is present in fish and seafood and considered not or less toxic to humans [6]. In contrast, the most prevalent toxic inorganic forms, arsenite and arsenate, are found in drinking water and food grown in arsenic-contaminated areas, such as rice [1,7]. Previous studies reported that exposure to inorganic arsenic causes oxidative stress, inflammation, and epigenetic modifications, leading to increased susceptibility to many adverse health problems such as cancers, cardiovascular diseases, diabetes, and cognitive dysfunctions [8,9]. They are metabolized to monomethylarsonic acid (MMA) and dimethylarsinic acid (DMA) through methylation and eliminated through urine [10,11]. Although there are multiple sources of exposure to inorganic arsenic, drinking water is known to be the main source of exposure when the water arsenic level exceeds the current regulatory level of 10 ppb [12].

Exposure to inorganic arsenic through drinking water has been a major public health problem worldwide, with an estimated of 100–200 million people exposed, with American and Asian countries being the most prevalent locations, in which millions of people are consuming contaminated water with As concentrations greater than 50 µg/L, five times greater than the current WHO standard (10 µg/L) [5,13]. As a result of this exposure, people with greater susceptibility, such as children and pregnant women, have been experiencing negative consequences, since inorganic arsenic has been associated with adverse effects during pregnancy, such as spontaneous abortions, fetal death, impaired fetal growth [14,15,16], and postnatal implications, such as neurodevelopment problems, respiratory and digestive morbidity [17,18,19], as well as increased risks of diseases in later life in populations with early-life exposure to the metalloid [20,21,22]. Early life exposures are mediated by the passage of arsenic through the placenta, since elevated placental concentrations have been related to high concentrations in maternal and infants’ urine [23], or through the consumption of breast milk from mothers who had arsenic-contaminated drinking water [24,25,26].

In Mexico, arsenic concentrations in water range from 7 to 600 µg/L, which are generally beyond the established regulatory level in Mexico of 25 µg/L [27,28], corresponding to the recent studies showing the that average concentration of drinking water arsenic is 82 μg/L in the CL, northern Mexico [29]. Additionally, studies based on a cohort maintained in the CL proposed an inverse relationship between arsenic metabolism in pregnant women exposed to arsenic in drinking water and adverse birth outcomes, such as birth weight and gestational age [30]. However, there have been no studies comparing the exposure levels to this metalloid between the binomial (mother and neonate) through biomonitoring.

In order to identify an early arsenic-exposure route in the CL child population, we assessed the determination of total arsenic levels in different binomial samples including blood, urine, placenta, and breast milk, in addition to contrasting them with samples from a comparison population not belonging to the CL region, without arsenic exposure from drinking water.

## 2. Materials and Methods

### 2.1. Study Location and Subjects

The CL is located in the middle of the states of Durango and Coahuila in Northern Mexico. This semi-arid area is one of the hotspots of groundwater arsenic contamination in the world, and elevated levels of arsenic have been reported in this area. Our previous study reported high levels of drinking water arsenic in this region, ranging from 20.6 to 709.3 µg/L [31]. Therefore, the residents in this area were considered as potentially exposed to high concentrations of arsenic from their drinking water.

Our initial study was conducted to assess the arsenic levels in drinking water and breast milk samples collected from 75 mothers in the rural areas of seven municipalities of two states (Durango and Coahuila) within the CL, and 39 mothers as a non-exposed comparison group from the municipality of Saltillo in the State of Coahuila, located in a straight line approximately 257 km away from the region with the exposure.

Our main study was expanded to assess arsenic exposure in both mothers and their newborns in the region. Eighty-three pairs of pregnant women and their newborns were recruited as the exposure group from the rural areas of seven municipalities in two states (Durango and Coahuila) in the Comarca Lagunera Region (Figure 1). In addition, 14 pairs of pregnant women and their newborns were recruited as a non-exposed comparison group from the municipality of Saltillo in the State of Coahuila. The inclusion criteria for both groups included factors such as being clinically healthy without any pathologies related to exposure to high levels of arsenic, having lived in the regions described for a minimum period of one year, and not having ingested seafood 8 days before collecting samples.

### 2.2. Sample Collection

Maternal blood and urine samples were taken from the mother immediately before delivery, as well as from the umbilical cord blood at birth. Both samples were collected in EDTA tubes as an anticoagulant. For the placental samples, these were taken from the top of the umbilical cord and washed repeatedly with an isotonic saline solution until all the blood residue was removed. Then, they were placed on absorbent paper to remove moisture excess and subsequently crushed and stored at −20 °C until analysis. Neonatal urine samples were taken 3 and 4 days after delivery. Breast milk samples for both studies were collected within 15 days after delivery, with prior personal hygiene, using a manual pump. The samples were transported cold and stored at −20 °C until processing.

### 2.3. Arsenic Determination

All samples were processed by wet digestion by the modified Cox method [32]. Total arsenic concentrations in water and biological samples were determined using a PerkinElmer Flow Injection Hydride Generation Atomic Absorption Spectrophotometer (Model AnalystTM 200, Waltham, MA, USA). For QA/QC purposes, the US National Institute of Standards and Technologies (NIST, Gaithersburg, MD, USA)-certified Standard Reference Materials for water (SRM 1643c), urine (SRM 2670), and bovine liver standard (SRM 1557c) for the placenta and breast milk were used. The sodium arsenite was obtained from Sigma Chemical Co. (St Louis, MO, USA), and all other reagents were obtained from Baker (Mexico). Distilled, deionized water was used for all analytical work; glassware was soaked in 10% nitric acid, rinsed with double-distilled water, and dried before use. The recovery rates ranged from 90% to 110% with the coefficients of variation between 0.5% and 12% based on calibration curves from standard solutions spiked with 10, 20, and 40 ng of total As. The limit of detection was 2.7 µg/L. Urinary creatinine was analyzed by the Jaffe method [33].

### 2.4. Ethical Considerations

In this study, all participants signed an informed consent letter. In this document, participants received information on the objectives of the study, as well as the potential benefits and risks. The study was approved by the Bioethics Committee of the School of Medicine of the Autonomous University of Coahuila, Campus Torreon, Coahuila, Mexico (approval by Secretaría de Salud and Comisiόn Nacional de Bioética in Mexico No. CONBIOETICA07CEI00320131015). It was explained that the risks were minimal since the participating mothers were only subjected to the collection of one blood sample per venipuncture.

### 2.5. Statistical Analysis

For the description of the data, mean, standard deviation, and ranges were used. The arsenic detection rates in breast milk samples between the groups were compared using a chi-square test. The comparison of the total As concentrations in each medium between the groups was performed through paired t-tests. The level of significance was established as α = 0.05, a value of *p* < 0.05 was indicative of statistical significance, and a *p* < 0.01 was indicative of high statistical significance. The statistical package SPSS version 21 was used.

## 3. Results

The initial study revealed that the average concentrations of total As in the drinking water were 47.7 and 0.05 µg/L in the exposed and unexposed areas, respectively. Table 1 presents the arsenic levels in the breast milk samples collected from mothers in the initial study. The initial study showed that 33% of breast milk samples collected from 75 mothers in the exposed region had levels of arsenic greater than the limit of detection, 2.7 µg/L (mean 8.5, range ND–26.0 µg/L). In the unexposed region, 13% of the 39 samples had detectable arsenic levels (mean 5.3, range ND–7.3 µg/L). The rate of arsenic detection in breast milk samples in the exposed region was significantly higher than that in the unexposed region (*p* = 0.02).

In the main study, the arsenic levels in the maternal placenta, peripheral blood, urine, and breast milk were determined (Table 2). The average placental arsenic level was 7.80 µg/kg (range 0.3–33) in the exposed region, which was significantly higher than that in the non-exposed region. The average maternal urinary arsenic level was significantly higher in the exposed region (54.92 µg/L and 77.04 µg/g-Cr), compared to the non-exposed region (4.60 µg/L and 6.71 µg/g-Cr). The arsenic levels in maternal blood and breast milk samples were also higher in the exposed region, although they were not statistically significant.

The creatinine-adjusted urinary arsenic levels three days after birth were significantly higher in neonates in the exposed region (mean 107.92, range 15.8–671.8 µg/g-Cr), compared to those in the comparison region (mean 14.78, range 14.08–33.18 µg/g-Cr) (Figure 2). Average urinary arsenic levels 4 days after birth was 17.57 (range 12.47–22.67 µg/g-Cr) in neonates in the exposed region, while arsenic was not detected in neonates 4 days after birth in the non-exposed region. The blood arsenic levels were slightly higher in the exposed group, which was not statistically significant.

## 4. Discussion

Total arsenic levels in drinking water greater than the global standard level of 10 µg/L have been detected in multiple recent studies around the globe [34,35,36,37,38]. In this study, in the CL, a known hydroarsenicism region, high concentrations of arsenic in drinking water were detected with an average level of 47.7 µg/L, almost fivefold above the international standard level and two-fold above the permissible level established by Mexican regulations (25 µg/L). Similar levels have been reported by recent studies conducted in the CL in order to assess a relationship between genetic variability and arsenic metabolism with an average water arsenic level of 82 μg/L [29], as well as possible plasma biomarkers, in order to assess inorganic arsenic exposure with concentrations of 22.1 μg/L [39].

High levels of arsenic in drinking water have been intrinsically related to a high level of arsenic exposure and internal levels in the body, and these, in turn, have increased the risks of many negative health effects, principally in populations with chronic exposure [28], which is the case for the population included in this study. A relationship has been reported between early life exposure and the development of a variety of diseases, such as carcinogenesis, atherosclerosis, and respiratory diseases, and an increase in mortality from the same diseases in later life [40,41,42,43,44,45,46,47]. This means that early life is the principal vulnerable window of adverse health outcomes associated with environmental exposures [48,49].

A study conducted in the US evaluated the correlation between arsenic levels in mothers’ urine and placental arsenic levels and in infants’ [23]. In this study, arsenic concentrations in the placenta (0.76 ng/g) were positively associated with arsenic levels in household drinking water, maternal urine, and toenail, and infants’ toenail samples (averages 0.38 µg/L, 3.62 µg/L, 28 µg/g, and 68 µg/g), indicating that placental passage is a major exposure route in utero [23]. A study confirmed that 80% of arsenic undergoes transplacental transfer by diffusion and found a strong correlation between arsenic levels in umbilical cord blood and maternal urine and toenail [50,51]. A longitudinal study conducted in Bangladesh reported the medians of arsenic levels obtained in women and their children’s urine samples were 96 µg/L and 35 µg/L, with a median of 66 ug/L in drinking water, which showed similar levels of water and urinary arsenic to our study [52]. A US study found associations between maternal urinary arsenic levels and birth outcomes, such as a decrease in head circumference and low birth weight, and the average maternal urinary arsenic level was 3.4 ug/L, which is much lower than the levels found in our current study [53]. This means that the newborns in our study may experience more severe health outcomes, demonstrating the importance of our study to evaluate the current exposure, as well as the future follow-up and preparation of community-engaged intervention strategies. High As levels maintained by the neonates through 4 days after birth, considering the urinary elimination half times for arsenic, suggest that the arsenic levels in the neonates could be attributed to another arsenic source, such breast milk [54]. Blood samples obtained from both mother and neonate were extremely low in contrast with the other sample determinations, which is in agreement with the notion that blood is not a suitable medium to assess arsenic exposure [55].

Previous studies reported that breast milk did not contain a substantial quantity of inorganic arsenic, although mothers were exposed to high levels of arsenic [56,57,58]; however, there is still a lack of sufficient data. Our study provides additional evidence that the prevalence and levels of arsenic in breast milk samples were relatively low (33%), despite high arsenic exposure from drinking water and the significantly higher arsenic detection rates in the CL compared to the comparison area. Fangstrom et al., (2008) showed that most inorganic arsenic found in human breast milk was the trivalent form, and the higher efficiency of arsenic methylation during pregnancy and lactation led to faster excretion and very low level of trivalent inorganic arsenic, or levels below the limit of detection in mothers [13,59]. Our study found that the average arsenic levels were 47 µg/L in water and 4 µg/L in breast milk, indicating 10-times lower arsenic exposure in breast milk samples, suggesting that exclusive breastfeeding may protect infants from arsenic exposure compared to feeding with formula mixed with arsenic-contaminated water. This is consistent with previous studies claiming a protective effect of breastfeeding due to the low arsenic excretion in breast milk [60,61].

A limitation of the present study was the absence of surveys on other covariates, such as socioeconomic status and food consumption pattern. For example, food is another source of exposure. When arsenic levels in drinking water are less than 10 ppb, food highly affects arsenic exposure. However, previous studies found that when drinking water levels of arsenic are greater than 10 ppb, water is the dominant source of exposure [12], indicating that the exposure of subjects in our study is primarily affected by drinking water (mean 47.7 µg/L). Another limitation of our study is the non-determination of arsenic species. There is evidence that suggests varying levels of this species during fetal growth, finding a higher proportion of methylarsonic acid (MMA) during the second and third trimesters. Additionally, the mother’s methylation efficiency would be increased during pregnancy [61,62]. This determination could be particularly relevant because there exist variations in the association with health status of newborns and the rate of metabolism [63].

## 5. Conclusions

This study demonstrated high arsenic exposure in mothers and their neonates in the CL through monitoring multiple biological samples, which demonstrates a continuous hydroarcenisism problem in this area. These findings confirmed placenta and breast milk as routes of early arsenic exposure in the local population. In addition, our study suggested that breastfeeding may reduce arsenic exposure in infants in arsenic-contaminated areas. Further studies are necessary in order to establish future comorbidity associations later in life, adjusted by other arsenic exposure sources and covariates, as well as the provision of interventions to reduce the exposure.

## Figures and Tables

**Figure 1 ijerph-19-16232-f001:**
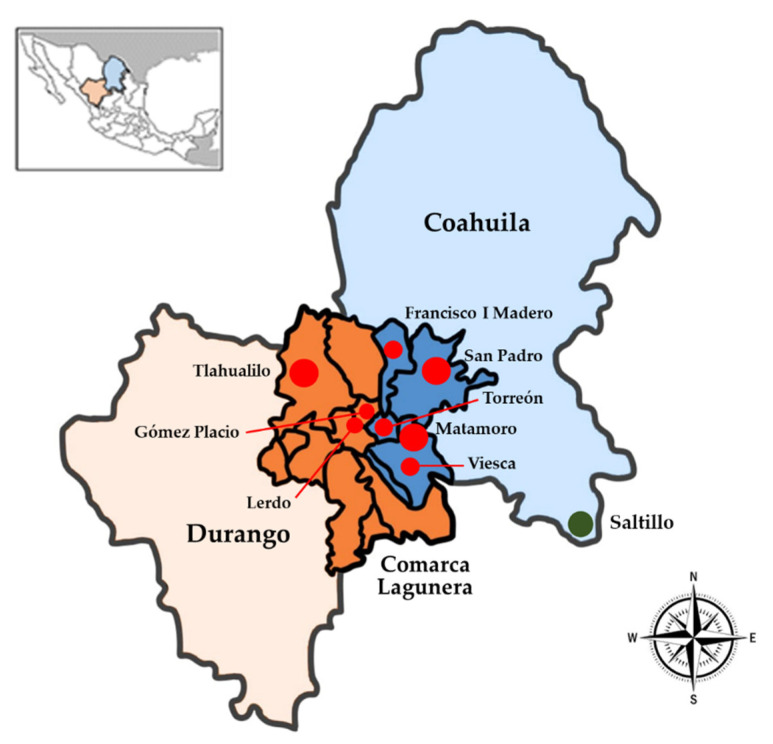
Geographical location of the Comarca Lagunera (CL). Dark orange and blue-colored areas indicate the CL region overlapping the Durango and Coahuila states, respectively. The red dots correspond to the municipalities of the Comarca Lagunera Region (exposed region). The green dot corresponds to the municipality of Saltillo, Coahuila (non-exposed region). Data taken from the National Institute of Statistics and Geography (INEGI, 2019).

**Figure 2 ijerph-19-16232-f002:**
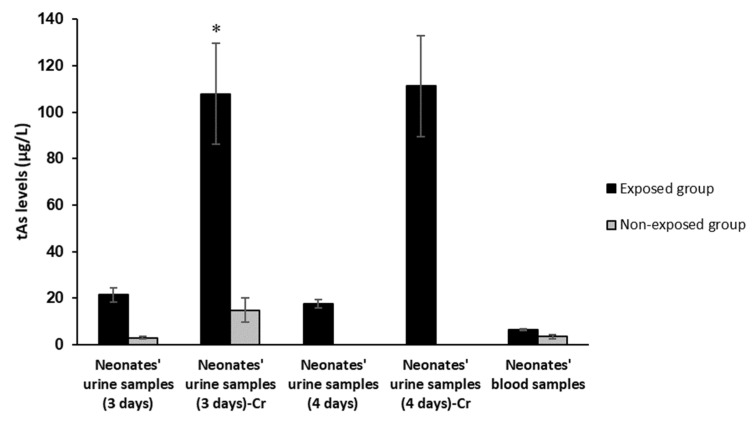
Total arsenic levels in newborns 3 and 4 days after birth in exposed and non-exposed regions (main study). Cr, levels adjusted by creatinine (µg of total As/g of creatinine). * *p* < 0.05.

**Table 1 ijerph-19-16232-t001:** Arsenic levels in maternal breast milk from the initial study.

Region	Total No.	No. with Detectable Arsenic in Breast Milk (%)	Mean (µg/L)	Range (µg/L)
Non-Exposed	39	5 (13)	5.3	ND ^1^–7.3
Exposed	75	25 (33) ^2^	8.5	ND–26.0

^1^ ND, non-detectable level. ^2^ Significantly higher detection rate than the non-exposed region based on a chi-square test (*p* = 0.02).

**Table 2 ijerph-19-16232-t002:** Arsenic levels in biological samples in mothers from the main study.

Medium	N	Mean	SD	Range
Exposed Region				
Placenta	83	7.80 µg/kg *	6.30	0.3–33
Blood	80	4.96 µg/L	2.90	ND–12.4
Urine	79	54.92 µg/L **	39.07	4.1–190
Urine-Cr	80	77.04 µg/g-Cr **	56.03	15.3–306.5
Breast Milk	75	4.30 µg/L *	10.50	ND–24.7
Non-Exposed Region				
Placenta	13	2.17 µg/kg	2.57	0.1–8.8
Blood	14	3.85 µg/L	2.64	ND–9.7
Urine	13	4.60 µg/L	3.09	0.8–9.4
Urine-Cr	13	6.71 µg/g-Cr	5.73	0.7–18.4
Breast Milk	13	0.87 µg/L	1.71	ND–7.4

Note: Cr, Adjusted by creatinine (µg of total As/gr of creatinine); SD, standard deviation; ND, non-detectable level. * *p* < 0.05; ** *p* < 0.001.

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
