# Peer review of "Comparative Biomonitoring of Arsenic Exposure in Mothers and Their Neonates in Comarca Lagunera, Mexico"

_ijerph, 2022, doi:10.3390/ijerph192316232_

Round 1
Reviewer 1 Report (Previous Reviewer 3)
No significant improvement from previous version has been made and the concerns raised before, such as lack of the novelty and the interpretation of table 1, still remain. The manuscript could be dramatically improved by including new information about the health outcomes of these subjects.
Author Response
Dear Reviewer 1, please find our answers in the attached PDF file.
We welcome your comments and contributions to this manuscript.

Reviewer 2 Report (New Reviewer)
The authors conducted a comparative analysis of arsenic levels in multiple biological samples from pregnant women and their neonates in CL and the comparison region, Saltillo. Total arsenic levels in placenta, breast milk, blood, and urine were measured in pregnant women and their neonates from rural area of seven municipalities of the CL using atomic absorption spectrophotometry with hydride generation methodology. They found The average concentrations of tAs in drinking water were 47.7 µg/L and 0.05 µg/L in the exposed and non-exposed areas. Mean levels of tAs were 7.80 µg/kg, 77.04 µg/g-Cr, 4.30 µg/L in placenta, blood, urine, and breast milk respectively in mothers and 107.92 ug/g-Cr in neonates in the exposed group, which were significantly higher than those in the non-exposed area. High levels of urinary arsenic in neonates maintained 4 days after birth. This is very interesting study, my specific comments are as follows: 1. If they can collected more samples, the result may make more sense; 2. They are many Arsenic species, why did author not detect the arsenic speciation? As we known, Arsenic with different chemical form has various toxicity. 3. Authors could extend the introduction section on the toxicity of As with different chemical forms; 4. More scientific explanation should be added for their results. 5. QA/QC section should be added.
Author Response
Dear Reviewer 2, please find our answers in the attached PDF file.
We welcome your comments and contributions to this manuscript.

Round 2
Reviewer 1 Report (Previous Reviewer 3)
Some new information has been added and the novelty of the manuscript has been improved.
This manuscript is a resubmission of an earlier submission. The following is a list of the peer review reports and author responses from that submission.
Round 1
Reviewer 1 Report
- No drinking water As concentration was presented, except a value in page 4. So, I don’t know how authors define the exposed or non-exposed region or population. In addition, water is not the only resource of As exposure.
- Statistical methods were misused. Did the concentration of As have a normal distribution? Mostly, levels of pollutants in fluid are not normal distributed. Additionally, correlation between mothers and neonates in As level should be analyzed. And pair-t test should be used in Figure3. Data presentation should be improved.
- How to confirm the sample size was not documented in the manuscript.
Reviewer 2 Report
This study conducted a comparative analysis of arsenic levels in different samples from pregnant women and their neonates in Mexico and demonstrated an early arsenic exposure route through placenta and breast milk. I think this is an interesting study with a good public significance, but I have several major concerns and this manuscript needs to be largely improved.
1. Many of the expressions throughout the manuscript is not in a usual style which is hard to understand, the authors should ask for a language editing service.
2. Methods: so many important contents are missing.
(1) The authors have to provide details on how they recruit the participants with specific inclusion and exclusion criteria.
(2) Also, a detailed description on the baseline survey relating to the study period, questionnaire and/or measurements should be provided.
(3) I think water intakes is a very important factor that could affect the arsenic exposure, but the authors did not report this data.
(4) The logic of the descriptions in section 2.1 is confusing. It is not easy to understand the difference between the pilot study and the main study. Please make it clear that does the numbers of 75 (exposed group) and 39 (non-exposed group) only represents the numbers of woman participants in the pilot study, and 83 (exposed group) and 14 (non-exposed group) are the numbers of mother – neonate pairs in the main study? But how to explain that maternal milk in the non-exposed group had 34 samples? This section has to be re-organized.
3. Results:
(1) Baseline demographic characteristics of the participants should be provided.
(2) Please have correlation and adjusted regression analyses concerning the associations of water intake, arsenic levels in mother’s samples with arsenic levels in the neonatal samples, which are crucial to answer the study question.
4. After the author address my concerns above, the Discussion section should be revised accordingly.
Reviewer 3 Report
The manuscript presented the results by comparing the arsenic content in placenta, blood, urine and breast milk samples of mothers/neonates between two populations (exposed group vs non-exposed group). The major issue is very limited new information presented. The authors reported that arsenic from foods/drinking water accumulated in the body, and may pass on from mothers to fetus through placenta and breast milk, which is well known. The other issue is the small sample size of the study. Further, in table 1, since arsenic is not detectable in 67% samples of exposed group and 87% samples of non-exposed group, not sure if there is a main effect of the arsenic in drinking water on the variables.